# A Review of Visual-LiDAR Fusion based Simultaneous Localization and Mapping

**DOI:** 10.3390/s20072068

**Published:** 2020-04-07

**Authors:** César Debeunne, Damien Vivet

**Affiliations:** ISAE-SUPAERO, Université de Toulouse, 10 avenue Edouard Belin-BP 54032, 31055 Toulouse, CEDEX 4, France; Cesar.DEBEUNNE@student.isae-supaero.fr

**Keywords:** SLAM, mapping, localization, hybridization, LiDAR, camera, vision

## Abstract

Autonomous navigation requires both a precise and robust mapping and localization solution. In this context, Simultaneous Localization and Mapping (SLAM) is a very well-suited solution. SLAM is used for many applications including mobile robotics, self-driving cars, unmanned aerial vehicles, or autonomous underwater vehicles. In these domains, both visual and visual-IMU SLAM are well studied, and improvements are regularly proposed in the literature. However, LiDAR-SLAM techniques seem to be relatively the same as ten or twenty years ago. Moreover, few research works focus on vision-LiDAR approaches, whereas such a fusion would have many advantages. Indeed, hybridized solutions offer improvements in the performance of SLAM, especially with respect to aggressive motion, lack of light, or lack of visual features. This study provides a comprehensive survey on visual-LiDAR SLAM. After a summary of the basic idea of SLAM and its implementation, we give a complete review of the state-of-the-art of SLAM research, focusing on solutions using vision, LiDAR, and a sensor fusion of both modalities.

## 1. Introduction

In the past few decades, autonomous navigation for mobile robots has been a very active research area. The main requirements for autonomous navigation are, first, a good and accurate localization of the robot itself and, second, good knowledge or perception of its environment. Today, the main localization systems used are the Global Navigation Satellite System (GNSS) solutions, which provide an absolute positioning on Earth with good precision. Nevertheless, such systems are not always available or accurate, depending the environment (tunnels, cave, city canyon, etc.) and can lead to errors of some meters, which is not acceptable for safe autonomous navigation. Moreover, a mobile robot needs to be able to navigate even in a dynamic environment with potential obstacles and does always not have any prior information about its environment (planetary exploration, search and rescue, etc.). The only way to make robots able to navigate is to represent the environment in some form. Generating a 3D map online appears to be the starting point for a complete autonomous navigation in a 3D world. Such a map can be composed of simple geometric features, but also of more complex semantic objects. With a consistent map, the robot will be able to detect free spaces, obstacles, and easily detectable landmarks in order to navigate precisely and safely. Doing so, the robot will be able to self-explore and map an unknown environment and to interact safely with it. The applications for such a mobile robot are numerous: space exploration, self-driving cars, subsea analysis, mining applications, search and rescue, structural inspection, and so on.

Such a navigation approach is called Simultaneous Localization and Mapping (SLAM). SLAM is the process by which a robotic system constructs a map of the environment using different kinds of sensors while estimating its own position in the environment simultaneously. Such a map can then be used by a human operator to visualize the environment and to set up the path of the robot or even by the robot itself to plan its own mission with autonomy. This is the case for autonomous navigation where the robot must plan its own path and make the right decisions without human intervention. Such a robot can maintain its own stability and plan its movement, even if some unexpected event occurs.

Currently, one of the most studied contexts for the SLAM application is the autonomous vehicle’s area, as it requires both localization and mapping to navigate in the environment. Recently, the industry has proposed cars called “autonomous”, but those cars are only a first step to autonomous navigation and must be considered as “semi-autonomous cars” as they only guarantee safe automated driving under very specific scenarios. Tesla autopilot guarantees at this time self-driving only on highway sections or in simple scenarios [1], but the entire attention of the driver is required. Considering the Google car, it is only designed to move itself on “wide open-roads” under good weather conditions with a preexisting accurate 3D map [2].

Most autonomous cars use Light Detection and Ranging (LiDAR) and/or stereo cameras to perceive the environment in which they are navigating. Such systems are very often hybridized with Differential GPS (D-GPS) or Satellite Based Augmentation Systems (SBAS) and Inertial Measurement Units (IMU) to robustify the localization solution [3]. With such sensors, if a very good GNSS is available, the localization accuracy can fall within some centimeters. However, in the case where GNSS cannot be trusted, other localization solutions have to be investigated. Most of the state-of-the-art techniques have attempted to solve this localization problem with exteroceptive sensors such as RADAR, LiDAR, and monocular/stereo cameras. By using a hybridization of such exteroceptive sensors with classical proprioceptive sensors (IMU, odometers), this allows reducing or removing the drift due to the cumulative error of such relative positioning approaches. It is interesting to note that the most used modalities (cameras and LiDARs) are two radically different sensors with their own strengths and weaknesses. For example, laser scanners are important for obstacle detection and tracking, but are sensitive to rain, while cameras are often used to get a semantic interpretation of the scene, but cannot work in bad lighting conditions. As they appear to be complementary, their fusion would permit balancing their main respective weaknesses. It is also interesting to note that the most challenging issue of the SLAM problem is the error accumulation that can reach arbitrarily high values [4]. Using both visual and LiDAR sensors may reduce the local uncertainties and then allow limiting the global drift.

The objective of this paper is to provide an overview of the existing SLAM approaches with a focus on novel hybridized LiDAR-camera solutions. In order to make this paper accessible to new researchers on SLAM, we will first present in Section 2 a short reminder about the theory behind the SLAM process. Then, because the main current state-of-the-art LiDAR-camera solutions are a simple concatenation of a visual-SLAM and a LiDAR SLAM, it is in our mind important to have an overview of SLAM for each modality. Section 3 will focus on the different kinds of visual-SLAM (V-SLAM) approaches, which means V-SLAM with a monocular and a stereo camera, but also modern RGB-D and event cameras. Then, Section 4 will give an overview of LiDAR based SLAM. Finally, in Section 5, we will discuss the state-of-the-art concerning the hybridized camera-LiDAR SLAM to understand the ground already covered and, in Section 6, what remains to be done.

## 2. Simultaneous Localization and Mapping

### 2.1. Principle

SLAM is a technique for estimating simultaneously the sensor motion and reconstructing the geometrical structure of the visited area. This technique was first developed in 1985 to achieve autonomous control of robots [5]. SLAM has been widely studied and applied with various kinds of sensors and for a multiple kinds of robotic platforms. Since 1985, there has not a month without a new communication about SLAM. Today, we begin to see some mature algorithms, but such approaches are still very dependent on the platform, on the environment, and on the parameters that have to be tuned.

The fundamental idea is that landmark correlations are used and are needed to improve the solution [6]. Combined with data association, SLAM solutions can perform loop closure to reduce the uncertainties of every pose of the map as every landmark in the loop appears to be correlated [4].

SLAM is an estimation problem. We want to estimate both a variable X that includes the robot trajectory or pose and a variable M representing the position of landmarks in the environment. Given a set of measurements $Z=\left\{z_{1},\ldots ,z_{m}\right\}$ and a measurement or observation model h(.) expressed $z_{k}$ as a function of X and M such as:(1)$$z_{k}=h\left(X_{k},M_{k}\right)+\epsilon_{k}$$
with $X_{k}$, $M_{k}$ respectively a subset of X and M and $\epsilon_{k}$ a random measurement noise. SLAM tends to solve the Maximum A Posteriori (MAP) problem in such a way that:(2)$$\begin{array}{cc} \left\{X^{*},M^{*}\right\} & =\underset{\left\{X,M\right\}}{\mathrm{argmax}}p\left(X,M|Z\right)\\ \; & =\underset{\left\{X,M\right\}}{\mathrm{argmax}}p\left(Z|X,M\right)p\left(X,M\right) \end{array}$$
p(Z|X,M) is the likelihood of the measurement Z given X and M, and p(X,M) is a priori knowledge of X and M.

Assuming that observations $z_{k}$ are independent, the MAP problem becomes:(3)$$\begin{array}{cc} \left\{X^{*},M^{*}\right\} & =\underset{\left\{X,M\right\}}{\mathrm{argmax}}\prod_{k=1}^{m}p\left(z_{k}|X,M\right)p\left(X,M\right)\\ \; & =\underset{\left\{X,M\right\}}{\mathrm{argmax}}\prod_{k=1}^{m}p\left(z_{k}|X_{k},M_{k}\right)p\left(X,M\right) \end{array}$$

This SLAM MAP problem was originally solved thanks to the Extended Kalman Filter (EKF) [7]. It reduces the uncertainties and gives an estimation at each step of the algorithm. Using a probabilistic model, the EKF guarantees convergence and the consistency of the map. However, it is very sensitive to data association errors, and the permanent update of all landmarks and their covariance matrix requires much computational effort. Figure 1 gives a block diagram of the EKF-SLAM process. Current state-of-the-art approaches solve this MAP problem thanks to optimization techniques such as Bundle-Adjustment (BA) [8] or deep neural network approaches [9,10].

### 2.2. Probabilistic Solution of the SLAM Framework

As seen before, SLAM is a recursive estimation process. Such a process is often viewed in a probabilistic way in which the classical prediction and update step must be fulfilled.

Considering a robot moving through an unknown environment, we define:$x_{k}$: the state vector describing the robot at time *k*$x_{k|k-1}$: the estimated state vector at time *k* given the knowledge of the previous state$u_{k}$: the control vector applied at k−1 to move the vehicle to a state $x_{k}$ (if provided)$m_{i}$: a vector describing the $i^{\mathrm{th}}$ landmark$z_{k,i}$: an observation of the $i^{\mathrm{th}}$ landmark taken at time *k**X*: the set of vehicle locations from Time 0 to *k*$U_{0:k}$: the set of control inputs from Time 0 to *k*$Z_{0:k}$: the set of observations from Time 0 to *k**M*: the set of landmarks or maps$M_{k|k-1}$: the estimated map at time *k* given the knowledge of the previous map at time k−1.

As we consider the probabilistic form of SLAM, at each time *k*, we want to compute the probability distribution function:(4)$$P(x_{k},M|Z_{0:k},U_{0:k})$$

To proceed, we have to use a recursive solution that uses the a priori $P(x_{k-1|k-1},M_{k-1|k-1}|Z_{0:k-1},U_{0:k-1})$ being updated by $u_{k}$ and $z_{k}$.

To do so, we first need to define a motion model providing a prediction of the state given the control input $P(x_{k}|x_{k-1},u_{k})$ in such a way that:(5)$$\begin{array}{c} P(x_{k|k-1},M_{k|k-1}|Z_{0:k-1},U_{0:k})=\ldots \\ \int P(x_{k|k-1}|x_{k-1|k-1},u_{k})\times \ldots \\ P\left(x_{k-1|k-1},M_{k-1|k-1}|Z_{0:k-1},U_{0:k-1}\right)dx_{k-1|k-1} \end{array}$$

In the same way, we also have to define a perception or observation model $P(z_{i,k}|X_{k},M)$ that will link the sensor data about detection *i* at time *k* to the estimated state in such a way that:(6)$$\begin{array}{c} P(x_{k|k},M_{k|k}|Z_{0:k},U_{0:k})\propto \ldots \\ P(z_{i,k}|x_{k|k-1},M_{k|k-1})\times \ldots \\ P(x_{k|k-1},M_{k|k-1}|Z_{0:k-1},U_{0:k}) \end{array}$$

Figure 2 shows an illustration of the SLAM process that represents all the variables used in this section.

At this step, we have computed both the prediction and update step, and SLAM can be presented as an iterative estimation respectively for the following equation, which is a combination of Equations (Equation 5) and (Equation 6):(7)$$\begin{array}{c} P(x_{k|k},M_{k|k}|Z_{0:k},U_{0:k})\propto \ldots \\ P(z_{k,i}|x_{k|k-1},M_{k|k-1})\times \ldots \\ \int P(x_{k|k-1}|x_{k-1|k-1},u_{k})\times \ldots \\ P\left(x_{k-1|k-1},M_{k-1|k-1}|Z_{0:k-1},U_{0:k-1}\right)dx_{k-1|k-1} \end{array}$$

Equation (Equation 7) gives the recursive Bayesian method for SLAM implementation. A solution to the SLAM problem must process an appropriate computation for both the motion model and perception model to compute the recursive method efficiently. Current state-of-the-art approaches commonly use inertial measurement unit mechanization as a prediction step or an assumption about the movement of the vehicle (constant velocity, constant acceleration, etc.). Considering the observation model, in the case of visual-SLAM, it is often based on the inverse depth or classical perspective view model. Considering LiDAR, RGB-D, or RADAR approaches, the observation model is much easier because the observation is a direct 3D measurement of the 3D world.
(8)$$P2D=\Pi \left(P3D,K,T\right)$$
with T=[R,t] the rigid transform providing the sensor’s 6D pose, *K* the intrinsics of the sensor, and Π(.) the perspective projection function. As can be seen, this function should be reversed to match to the classical observation model (P3D=g(P2D,K,T)). However, such inversion is not straightforward, so the estimation step is often postponed for an additional P2D observation. Then, $P3D_{map}$ triangulation is processed.

### 2.3. Graph Based Solution of the SLAM Framework

In the LiDAR case, the model is straightforward as the observation is directly linked to the state by a classical rigid transform such as:(9)$$P3D_{map}=T\left(P3D_{LiDAR}\right)$$

Even if a probabilistic framework can be followed with LiDAR data, such SLAM approaches are often solved thanks to graph based methods in which only the collection of relative transformation is optimized [11].

The SLAM graph based formulation was proposed by [12]. It constructs a simpler estimation problem by abstracting the raw sensor measurements. The raw measurements are replaced by the edges in the graph, which can be seen as “virtual measurements”. In fact, such an edge is labeled with a probability distribution over the relative locations of the two poses conditioned on their mutual measurements. As seen in Figure 3, the process is composed of two main blocks: graph construction (front-end) and graph optimization (back-end). Most of the optimization techniques focus on computing the best map given the constraints: these are the SLAM back-ends. In contrast, SLAM front-ends seek to estimate the best constraints from the sensor data.

For a tutorial about graph based SLAM, one can refer to [13].

## 3. Visual-SLAM

After this reminder about SLAM theory, the objective of this section is to present a brief overview of all the existing approaches to perform visual-SLAM. Visual-SLAM is one of the most active research areas in robotics. Visual sensors had been the main research direction for SLAM solutions because they are inexpensive, capable of collecting a large amount of information, and offer a large measurement range. The principle of visual-SLAM is quite easy to understand. The objective of such a system is to estimate sequentially the camera motions depending on the perceived movements of pixels in the image sequence. This can be done in different ways. A first approach is to detect and track some important points in the image; this is what we call feature based visual-SLAM. Another one is to use the entire image without extracting features; such approach is called direct SLAM. Of course, also other SLAM solutions exist using different cameras such as RGB-D or Time-of-Flight (ToF) cameras (which provide not only an image, but also the depth of the scene), but also event cameras (detecting only changes in the image).

In this section, we propose, for clarity, decomposing visual-SLAM into these different families.

### 3.1. Feature Based SLAM

Feature based SLAM can be decomposed again into two sub-families: filter based and Bundle Adjustment based (BA) methods. A comparison between such approaches was proposed in [14]. The first monocular approach MonoSLAM was proposed in 2003 by Davison et al. [15,16]. The estimation of both features and pose was simply done by using an extended Kalman filter. Such filter based techniques have been shown to be limited in the case of a large environment as too many features have to be saved in the state. In order to reduce this problem, PTAM [17] was proposed in 2007. It splits the pose and map estimation into different threads and proposes to use BA. Of course, many extensions have also been proposed [18,19]. In order to improve feature based SLAM with BA, loop closing had been added in order to detect if a keyframe has already been seen [20,21]. At the time of the writing of this article, the most used algorithm for SLAM is ORB-SLAM [22,23]. Such an algorithm includes most of the “tricks” to improve SLAM’s performances and can handle monocular, stereo, and RGB-D camera configurations with different algorithms. The problem of such algorithms is the great number of input parameters to tune in order to make the SLAM work in a given environment. However, if we know enough about the environment to tune every thing, do we really need to perform SLAM anymore? The work proposed in [24] tried to reduce this number of parameters and to make visual-SLAM independent of the platform and environment, but did not reach the performances of ORB-SLAM.

### 3.2. Direct SLAM

In contrast to feature based methods, direct methods directly use the image without any feature detectors and descriptors. Such feature-less approaches use in general photometric consistency to register two successive images (for feature based approaches, the registration is based on the geometric positions of feature points). In this category, the most known methods are DTAM [25], LSD-SLAM [26], SVO [27], or DSO [28,29]. Finally, with the development of deep learning, some SLAM applications emerged to imitate the previous proposed approaches [9,30]. Such research works generated semi-dense maps representing the environment, but direct SLAM approaches are time consuming and often require GPU based processing.

### 3.3. RGB-D SLAM

The structured light based RGB-D camera sensors [31,32] recently became inexpensive and small. Such cameras can provide 3D information in real time, but are most likely used for indoor navigation as the range is inferior to four or five meters and the technology is very sensitive to sunlight. One can refer to [33,34,35,36,37,38] for RGB-D vSLAM approaches.

### 3.4. Event Camera SLAM

Finally, an event camera is a bio-inspired imaging sensor that can provide an “infinite” frame rate by detecting the visual “events”, i.e., the variations in the image. Such sensors have been recently used for V-SLAM [39,40,41]. Nevertheless, this technology is not mature enough to be able to conclude about its performances for SLAM applications.

### 3.5. Visual SLAM Conclusions

As you can see, the V-SLAM research area is very rich, and we just provided a review of the main approaches. For a more complete review of visual-SLAM, one can read [42,43]. Even if V-SLAM provides very good results, all of those V-SLAM solutions are prone to errors because of their sensitivity to light changes or a low textured environment. Moreover, RGB-D based approaches are very sensitive to daylight because they are based on IR light. As a result, they perform well only for indoor scenarios. Considering other visual approaches, they perform poorly in untextured or poor environments in which no pixel displacement can be estimated accurately. Finally, image analysis still requires high computational complexity. A summary can be found in Table 1.

Those drawbacks motivate researchers to create optimized and strong algorithms that can handle data errors and reduce execution time. For all of these reason, other sensors have also been investigated for the SLAM process. Currently, the first autonomous car prototypes mainly rely on other sensors: RADAR or LiDAR.

## 4. LiDAR Based SLAM

The common point to every mobile robot designed to perform SLAM is that they all use exteroceptive sensors. Even if RADAR based SLAM proved to be efficient [44,45,46], we choose to focus our attention in this paper on laser scanning devices. One reason for this choice is that RADAR is not accurate enough yet to provide a good 3D mapping of the surroundings of the vehicle, and so, its fusion with visual sensors is quite difficult. Considering LiDAR, 3D mapping with laser scanners remains a popular technology because of the simplicity, but also because of the accuracy. Indeed, applied to the SLAM problem, LiDAR can achieve a low drift motion estimation for an acceptable computational complexity [47].

Laser scanning methods appear to be a cornerstone to both the 2D and 3D mapping research. LiDAR can deliver point clouds that can be easily interpreted to perform SLAM. Stop-and-scan [48] was one of the first attempts to reach a proper SLAM solution with LiDAR. It avoids motion distortion, but is not a reliable solution for navigation purposes. The fusion with IMU can correct motion distortion using an error model that takes the velocity information as input [49]. While IMU is often used to undistort the data, it is also often used to predict the motion. The work in [50] showed that such a method can lead to over-convergence and proposed odometry based only on LiDAR distortion analysis.

It is interesting to note that even if LiDAR’s application is huge, the technique used for LiDAR scan registration has remained the same for almost ten years. The main solution for LiDAR based navigation is the scan-matching approach followed by a graph optimization.

### 4.1. Scan-Matching and Graph Optimization

Scan-matching is the fundamental process used to create 3D maps with LiDAR and gives precise information on motion. The general approach for registering 3D point clouds is the Iterative Closest Point (ICP) [51]. For the principle, see Figure 4. Its main drawback is the expensive search for point correspondences and its great sensitivity to the minimization starting point. To tackle this issue, kd-tree structures [47] can be introduced to accelerate the search for the closest point. The work in [52] showed that ICP robustness can be enhanced by using a probabilistic framework that takes into account planar structures of scans; this is generalized ICP. An alternative method is a Polar Scan Matching (PSM) [53] that takes advantage of the polar coordinates delivered by the laser scanner to estimate a match between each point.

In order to reduce local errors, graph based methods [54] can be used with LiDAR. The history of robot poses is represented by a graph: every node represents a sensor measurement, and the edges represent the constraint generated by the observation (coming from the ICP result). All method, relying on the pose graph, can be solved using various optimization methods like for example the Levenberg–Marquardt optimizer.

As an example from aircraft navigation, the use of 2D LiDAR combined with GNSS and IMU was proposed in [55].

Let us note that scan-matching can be processed for both 2D and 3D LiDAR. Considering 2D LiDAR application, filtering based approaches have also been proposed in the case of a “flat” world assumption.

#### 4.1.1. Occupancy Map and Particle Filter

Another effective means to solve the SLAM problem is the use of a Rao Blackwellized particle filter like Gmapping [56]. It greatly reduces the local errors and offers interesting results on planar environments. Each particle represents a possible robot pose and map. However, the big number of particles required to map the environment correctly can lead to non-negligible computation time. The work in [57] showed a particle filter applied to 2D SLAM able to compute a highly accurate proposal distribution based on a likelihood model. The result was an accurate occupancy grid map obtained with a number of particles one order of magnitude smaller compared to classical approaches. Of course, adapting such techniques to 3D is very difficult due to the size of the occupancy grid.

#### 4.1.2. Loop Closure Refinement Step

Previous solutions allowed obtaining a localization and built a map of the environment in an odometric way. To answer the SLAM problem fully, the loop closing step has been added to LiDAR odometry. In order to improve the global map consistency, loop closure can be performed when the robot locates itself in a pre-visited place. It can be performed thanks to feature based methods like [58]. For laser scans, geometric descriptors are used like lines, planes, or spheres. Those descriptors are used to perform matching between scans to detect an eventual loop. As a scan matcher between every scan can hardly run in real time, submaps that represent the environment of several scans were used in [59]. All finished submaps are automatically inserted in the scan matching loop that will make the loop detection on a sliding window around the current robot pose. Magnusson et al. [60] proposed an original loop detection process using the Normal Distributions Transform (NDT) representation of the 3D clouds. It is based on feature histograms that describe the surface orientation and smoothness.

The work in [55] demonstrated how efficiently the global drift of a LiDAR-SLAM could be corrected by performing loop closure. In their case, the Kalman filter was simply augmented with a place recognition module able to detect loops. A summary of LiDAR based SLAM is presented in Table 2.

## 5. LiDAR-Camera Fusion

As seen before, SLAM can be performed both thanks to visual sensors or LiDAR. Visual sensors have the advantage of being very well studied at this time. Even if V-SLAM provides accurate results, there are some defaults like: the drift of the scale factor in the monocular case, the poor depth estimation (delayed depth initialization) or the small range for stereo-vision, the sparsity of the reconstructed maps (for feature based indirect approaches), the difficulty to use RGB-D in outdoor scenarios, etc. Considering the 3D LiDAR based SLAM, the techniques used rely on scan matching and graph pose. Some solutions focus on landmark detection and extraction, but the point clouds obtained are usually not dense enough to be efficient. Nevertheless, the main advantage of LiDAR is its very good accuracy in ranging and, as a consequence, in mapping. Today, it is evident that the fusion of both modalities would be of great help for modern SLAM applications. Of course, using both modalities requires a first difficult calibration step. This section will present both the calibration tools available and the current state-of-the-art of LiDAR-camera fusion approaches.

### 5.1. The Mandatory Calibration Step

In order to perform SLAM with a LiDAR-camera fusion with optimal performance, precise calibration must be guaranteed between the two sensors. An extrinsic calibration is needed to determine the relative transformation between the camera and the LiDAR, as is pictured in Figure 5.

One of the first toolboxes to propose an interactive solution to calibrate a camera to LiDAR was [61]. It consists of manually marking the corresponding points of both the LiDAR scan and camera frame. The work in [62] details an automatic camera-laser calibration using a chessboard. It performs straight line extraction in order to deduce a proper rigid transformation between the two sensors. However, those offline calibration techniques cannot be used for optimal extrinsic calibration, as extrinsic parameters can change daily and require very specific conditions to work.

As deep Convolutional Neural Networks (CNN) have recently become popular in robotic applications, the work in [63] proposed a calibration based on CNN. The CNN takes the LiDAR and camera disparities as inputs and returns calibration parameters. This gives a fast online calibration solution that is suitable for real-time applications.

At this time, there is still no solution commonly used to process such a calibration in a simple but accurate way.

### 5.2. Visual-LiDAR SLAM

#### 5.2.1. EKF Hybridized SLAM

In the context of visual-LiDAR SLAM, it has been proven that the classical formulation of Extended Kalman Filter (EKF) SLAM can be modified to integrate such sensor fusion. The work in [64] proposed a new expression of EKF using data association, leading to an improvement in SLAM accuracy. The work in [65] also offered an RGB-D camera with LiDAR EKF SLAM. The main purpose of this work was to tackle the issue of unsuccessful visual tracking. If it fails, the LiDAR pose is used to localize the point cloud data of RGB-D camera to build a 3D map. Such an approach does not really provide a fusion, but a switch mechanism between the two modalities. The work in [66] integrated different state-of-the-art SLAM algorithms based on vision and inertial measurement using EKF on a low cost hardware environment for micro aerial vehicles. A 2D LiDAR was incorporated into the SLAM system to generate a 2.5D map and improve the robot pose estimation. Such proposed approaches still are loose coupling approaches that do not rely on feature detection on the measurement space. More tightly coupled LiDAR-vision sensor fusion algorithms are still missing in the literature.

#### 5.2.2. Improved Visual SLAM

From another point of view, the great performances reached by visual-SLAM algorithms motivated the use of sensor fusion to get optimal solutions on those frameworks. In [67], LiDAR measurement was used for depth extraction. After projection of the point cloud on the frame, motion estimation and mapping were performed using a visual keyframe based bundle adjustment. The work in [68] proposed a direct visual-SLAM using a sparse depth point cloud from LiDAR (Figure 6). However, as camera resolution is much higher than LiDAR resolution, a great number of pixels do not have depth information. The work proposed in [69] gave a solution to tackle the issue of resolution matching. After computing the geometrical transformation between the two sensors, a Gaussian process regression was performed to interpolate the missing values. As a result, LiDAR was used just to initialize the features detected in the images directly in the same way as with RGB-D sensors.

Zhang et al. [70] proposed a monocular SLAM associated with a 1D laser range finder. As monocular SLAM often suffers from scale drift, this solution gave an efficient drift correction for a very low hardware cost. Scherer et al. [71] mapped the course and vegetation along a river thanks to a flying robot and a hybridized framework. The state estimation was performed by visual odometry combined with inertial measurement, and LiDAR was used for sensing obstacles and mapping the river borders. However, point clouds could contain occluded points that deteriorate the estimation accuracy. The work in [72] tackled this issue by proposing a direct SLAM method with an occlusion point detector and a co-planar point detector. In these last articles, the visual-SLAM estimated pose was used to register the LiDAR point cloud during the mapping phase.

#### 5.2.3. Improved LiDAR SLAM

In many cases of visual-LiDAR SLAM, LiDAR is used for motion estimation through scan-matching, while the camera performs feature detection. Liang et al. [73] enhanced the poor performance of a LiDAR based SLAM using scan matching with a visual loop detection scheme using ORB features. In [74], a 3D laser based SLAM was associated with a visual method to perform loop detection through a keyframe based technique using visual bags-of-words. Furthermore, Iterative Closest Point (ICP) can be optimized using LiDAR-camera fusion. The work in [75] used visual information to make an initial guess for rigid transformation that was used to seed a generalized ICP framework.

#### 5.2.4. Concurrent LiDAR-Visual SLAM

Other works tried to combine both LiDAR and visual-SLAM results. The work in [76] proposed to use both visual and LiDAR measurements by running in parallel SLAM for each modality and coupling the data. This was done by using both modalities’ residuals during the optimization phase. Zhang et al. [77] combined their previous works to design VLOAM. This visual-LiDAR odometry performs high frequency visual odometry and low frequency LiDAR odometry to refine the motion estimate and correct the drift.

Maybe the most tight fusion currently available was proposed in [78], where a graph optimization was performed, using a specific cost function considering both laser and feature constraints. Here, both the laser data and image data could obtain the robot pose estimation. A 2.5D map was also built to accelerate loop detection.

### 5.3. Summary

To sum up, these examples mainly used sensor fusion to give additional information to a LiDAR-only or visual-only SLAM framework. Among all the ways of implementing such a SLAM (see Figure 7), the hybridized framework is the least studied. Creating a common SLAM framework using visual information and laser range seems to be a real challenge. A more tightly coupled LiDAR-vision sensor fusion algorithm is not fully investigated in the literature yet and should be studied.

## 6. Discussion on Future Research Directions

After this review of the literature, it appears that a complete tightly fused visual-LiDAR approach taking the advantages of both sensor modalities does not exist yet. We state that using LiDAR features as visual features in a tight hybridized fashion would benefit the SLAM community. Indeed, solving a multi-modal, hybrid multi-constraint MAP problem would be feasible. Such a solution would make SLAM more robust to environmental conditions such as light or weather. It is well known that V-SLAM does not work in poor lighting conditions or texture-less environments, but LiDAR SLAM can. On the other hand, LiDAR-SLAM performs poorly in rainy conditions (detection of the wrong impacts) or in textured, but not geometrically salient areas (open field, very long corridor) where camera based SLAM works perfectly.

We propose to investigate some hybrid approaches using a set of different extracted landmarks coming from different modalities such as $L=\{L_{vision},L_{LiDAR}\}$ in a multi-constraint MAP approach (see Figure 8).

The proposed framework follows a classical SLAM architecture (as we proposed in [24]) with three main steps: (1) the data processing step, which performs feature detection and tracking on both modalities; (2) the estimation step, which first estimates the vehicle displacement from the tracked features (this can be done by ICP, epipolar geometry, proprioceptive sensors, or a fusion of each. for example a Kalman filter or a multi-criteria optimization), then tries to detect and match landmarks from the map to the features; once matching is done, the pose can be refined (filtering/optimization), and finally, new landmarks can be estimated; the final step (3) deals with the global mapping: is the current data defining a keyframe (does it bring enough new information), and depending on the detection of a loop closing, does it optimize the trajectory locally or globally?

For this framework to work, efforts must be mainly made on (1) the LiDAR scan analysis for feature detection and (2) the camera-LiDAR calibration process. Once accurate solutions are found for these two problems, a tight coupling between LiDAR and vision will be possible at the data level, making the state estimation much more accurate.

## 7. Conclusions

Various studies have been conducted by researchers to find the best implementation of SLAM. As it is proven that it is possible for an autonomous robot to estimate its own pose and the map of its surroundings concurrently, SLAM remains a promising and exciting research subject in robotics. Theoretically, it is a complete solution to autonomous navigation; however, in practice, numerous issues occurs. Even if it appears to be a very promising solution, can we predict how far the development of SLAM can lead to real autonomous navigation? Therefore, it is necessary to deepen our understanding regarding SLAM and its contribution to the artificial intelligence mobile robot.

At this time, some robust and efficient solutions exist using visual sensors hybridized with the IMU. Such approaches are used today in industrial application mainly based on virtual or augmented reality. RGB-D cameras are a hot topic, but such sensors do not perform well in an outdoor environment where the ambient light greatly disturbs the detection. Visual approaches are prone to drift and are very sensitive to the lack of salient features in the environment. To overcome the drawback of the lack of features in indoor monotonic environments, geometrical features have been studied such as line, segment, or edgelets. The main problems of such landmarks are (1) the lack of accurate descriptors for the matching phase and (2) the difficult initialization phase of the corresponding 3D object with few detections. As a consequence, the 3D sparse representation of the environment is not very accurate due to mismatched features or wrong initialized ones. Finally, some hybrid maps are generated with different kinds of landmark representation. A generalized, multi-constraint MAP problem is then solved using these different objects and observations.

On the other hand, LiDAR based SLAM also exists and provides very good solutions. LiDAR approaches provide very accurate 3D information of the environment, but are often time consuming and still rely on very simple scan-matching approaches that are not very robust. At this time, very few works deal with an analysis of the 3D scan by extracting some 3D landmarks. None of the SLAM approaches using 3D LiDAR deal with landmarks in a similar way as in the vision based framework. The reason for this is the processing time required for the analyses and extraction of LiDAR landmarks. At this time, planes are the only features used in LiDAR-SLAM approaches. However, planes are not very useful in natural outdoor environments, which are by nature not well structured. LiDAR based SLAM is mainly based on scan-matching approaches such as ICP. Such algorithms have remained almost the same since its their invention thirty years ago.

Some experiments have been done to couple both LiDAR and visual sensors, but all of them remain at a very loose fusion level. The fusion is mainly done using the result of both odometry steps, which means that LiDAR detection or visual detection cannot help each other, and the decision is made at a very late step while fusing the relative displacement estimation. Other approaches only use the depth measurement of LiDAR to initialize the visual features directly. Once again, the capability of LiDAR is totally underused.

In our future work, we will study a tightly hybridized implementation of SLAM using sensor fusion. By fusing frames from a camera to point clouds from a LiDAR, we expect to build a robust and low-drift SLAM framework. Furthermore, as LiDAR prices are going down through the years, we expect that over time, such a solution will become low cost.

## Figures and Tables

**Figure 1 sensors-20-02068-f001:**
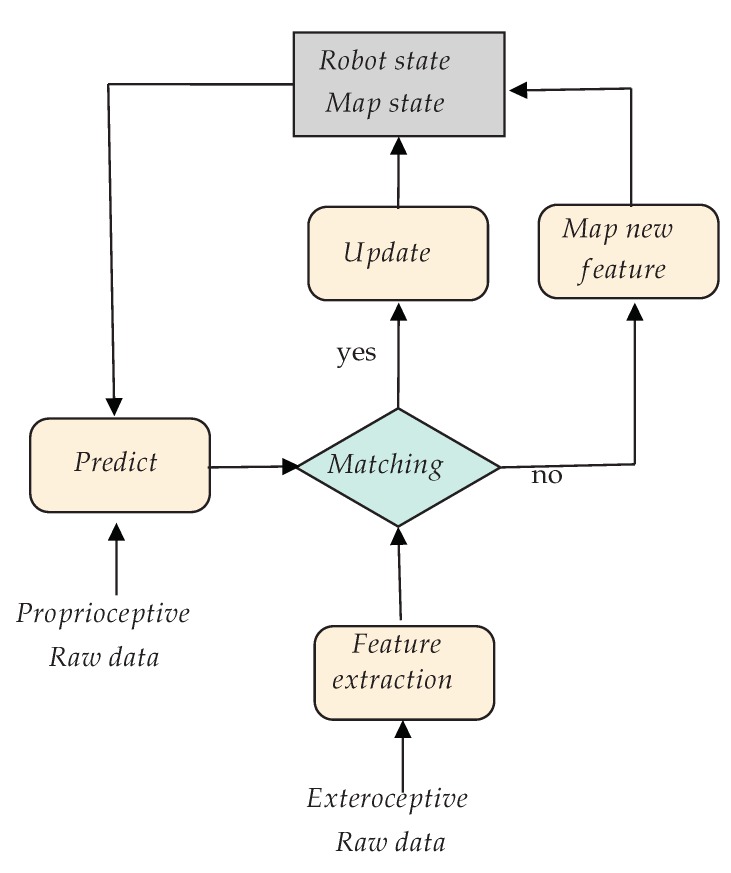
Block diagram of the EKF-SLAM process. When exteroceptive data are coming at time *t*, the state of the robot is predicted at this time with Equation (Equation 5), and the detected features are matched with those in the map. The matches allow updating the state and map using Equation (Equation 6). If the detection is not in the map, it is initialized when possible and added to the map. This process is done recursively (see Equation (Equation 7)).

**Figure 2 sensors-20-02068-f002:**
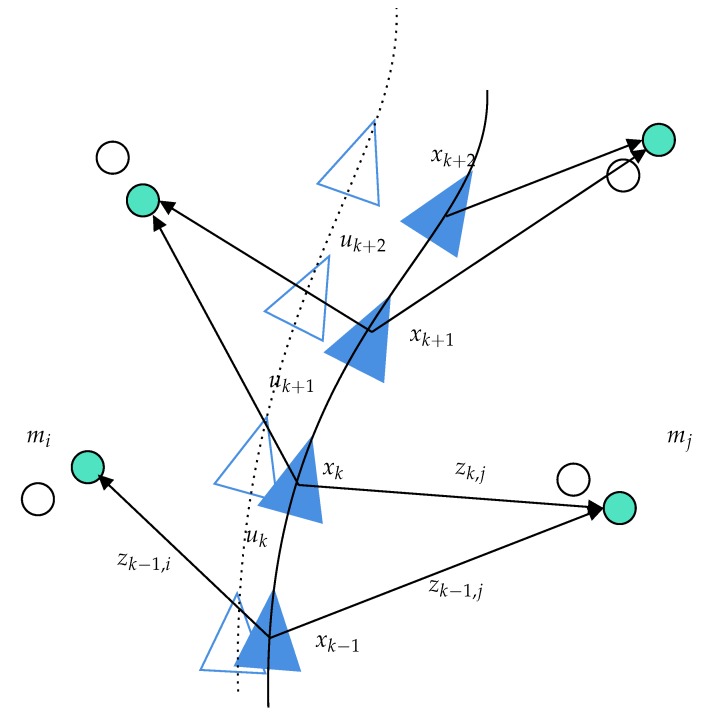
SLAM process illustration and notations. In blue, the estimated trajectory and map; in white, the ground truth.

**Figure 3 sensors-20-02068-f003:**
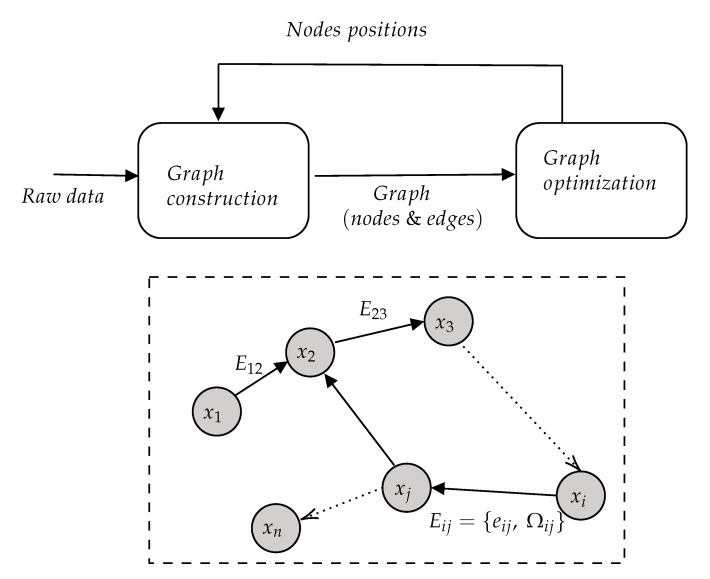
Representation of a pose-graph SLAM process. Every node $x_{i}$ in the graph corresponds to a robot pose. Nearby poses are connected by edges $E_{ij}=\left\{e_{ij},\Omega_{ij}\right\}$ that represent the spatial constraints between the robot poses *i* and *j* (coming from measurements). Note that edges $e_{ij}$ model the proprioceptive information constraint and $\Omega_{ij}$ the exteroceptive one.

**Figure 4 sensors-20-02068-f004:**
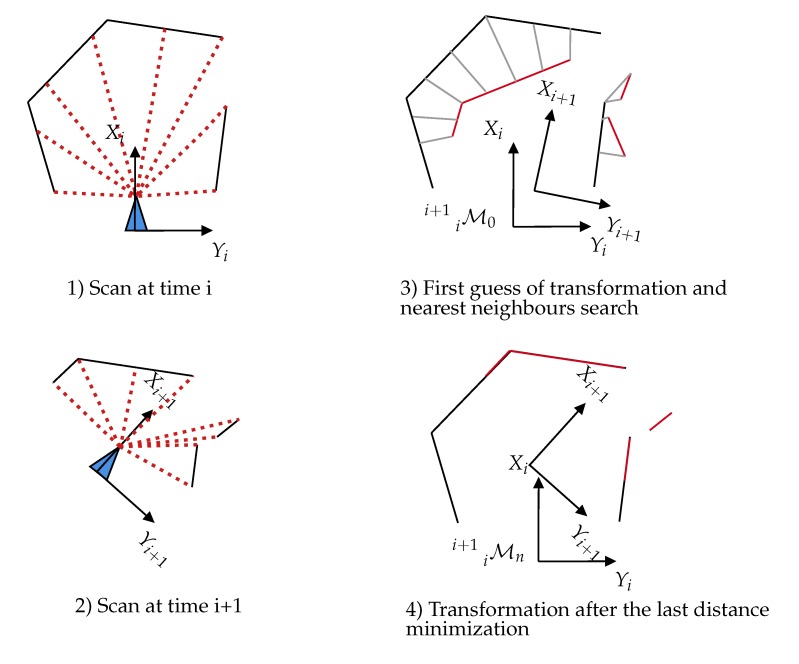
Principle of the ICP algorithm. At each iteration, the closest points are extracted between the two scans. From these matches, a transformation is processed and applied on the second scan. Then, the process iterates until a given cost criterion is reached.

**Figure 5 sensors-20-02068-f005:**
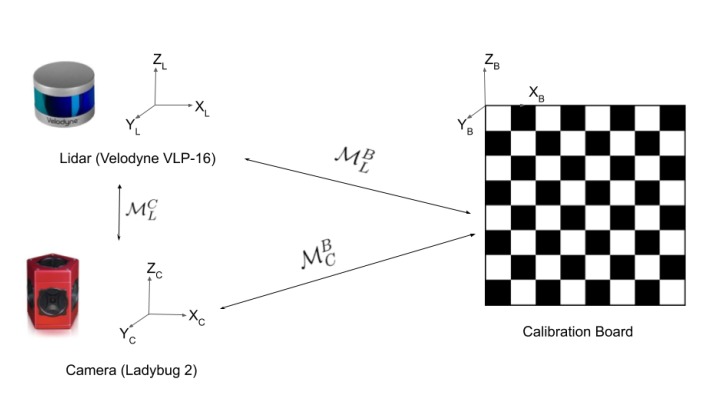
Principle of extrinsic calibration. The objective is to find the rigid transform $M_{L}^{C}$ between the LiDAR and the camera. It is currently mostly done manually using a calibration target like a 2D or 3D chessboard or pattern and by detecting this pattern for each modality ($M_{L}^{B}$ and $M_{C}^{B}$).

**Figure 6 sensors-20-02068-f006:**
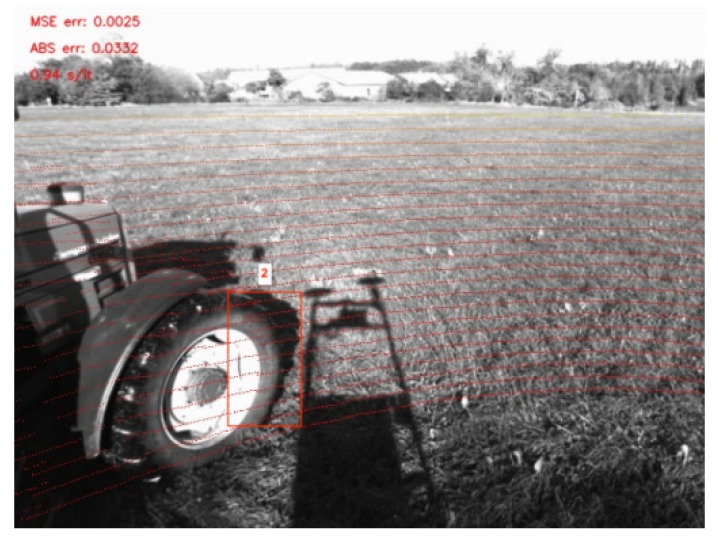
Example of LiDAR reprojected on the image. Note that a small error in the calibration will result in strong error on the estimated depth (see Square 2). We processed the error w.r.t. the depth obtained by stereo-vision.

**Figure 7 sensors-20-02068-f007:**
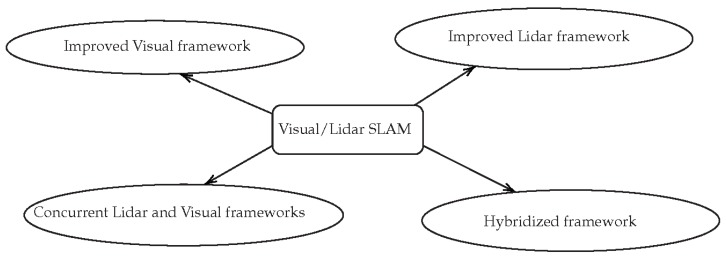
Different ways of implementing visual-LiDAR SLAM.

**Figure 8 sensors-20-02068-f008:**
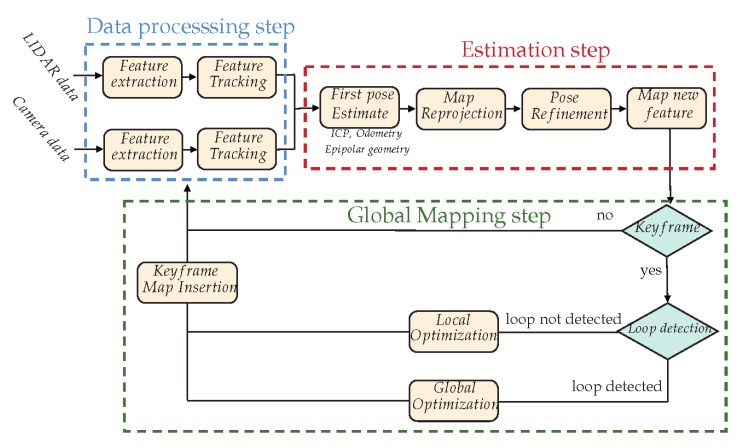
Proposed Hybridized SLAM global framework. The proposed framework follows a classical SLAM architecture with three main steps: (1) The data processing step, (2) the estimation and (3) the global mapping step.

**Table 1 sensors-20-02068-t001:** Summary of visual based SLAM’s advantages and drawbacks.

Visual Based SLAM				
	**Feature Based**	**Direct**	**RGB-D**	**Event-Based**
Advantages	Low SWAP-C (Low Size, Weight, Power, and Cost)	Semi-dense map, no feature detection needed	Very dense map, direct depth detection	“Infinite” frame-rate
Drawbacks	Sensitive to texture and light	Computational cost, required photometric calibration, often GPU based	Sensitive to daylight, work only indoor, very huge amount of data, very short range	Sensors expensive, detect only changes of the environment

**Table 2 sensors-20-02068-t002:** Summary of LiDAR based SLAM’s advantages and drawbacks.

LiDAR Based SLAM		
	**Occupancy Map**	**Graph-Based**
Advantages	Well known and easy to use, very accurate and precise, standard for 2D LiDAR	Allows large-scale SLAM, removes the raw data from the optimization step, allows a very simple loop closing process
Drawbacks	Dedicated to 2D environments only, cannot be used in 3D, very huge amount of memory required for large-scale environments, difficult loop closing process	Need to estimate very accurately the edges and the statistical links between the nodes; the mapping is an optional step

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
