# Peer review of "A Review of Visual-LiDAR Fusion based Simultaneous Localization and Mapping"

_sensors, 2020, doi:10.3390/s20072068_

Round 1

Reviewer 1 Report

The SLAM problem is a hot research area in autonomous robotics. In this paper, the Visual-LIDAR SLAM is reviewed. Generally, the paper is well written. The detailed comments are as follows:

  1. In equations (2) -(4),  what does P(xk|k-1,Mk|k-1|Z0:k-1,U0:k)  represents for? The definitions on xk|k-1, Mk|k-1 are not  given.
  2.  section 2.2. only reviewed Probabilistic solution of SLAM framework. What about the graph optimization framework? For completeness, it would be better is the authors give a berief discussion on it.
  3. Figure 9 shows the Hybridized SLAM global framework, it can be further improved and give more details.
  4. The authors are suggested to give more figures or tables for better undstanding and more clear orgnization and presetation of this survey paper.

Author Response

Thank you for your review. Please find our answers in the text :

In equations (2) -(4),  what does P(xk|k-1,Mk|k-1|Z0:k-1,U0:k)  represents for? The definitions on xk|k-1, Mk|k-1 are not  given.

Thank you highlighting this. We added the definitions in the text.

P (x, M | z, u) represents the probability density function of the state and map of the robot given the complete set of exteroceptive and proprioceptive data. It is in fact the desired output of the SLAM: the estimated robot state and map

section 2.2. only reviewed Probabilistic solution of SLAM framework. What about the graph optimization framework? For completeness, it would be better is the authors give a berief discussion on it.

You are totally right. We speak about it in the bibliography section but did not introduce it in the principle section. We added a subsection speaking about graph-based framework with some references.

Figure 9 shows the Hybridized SLAM global framework, it can be further improved and give more details.

We have modified the figure and caption to be more informative

The authors are suggested to give more figures or tables for better undstanding and more clear orgnization and presetation of this survey paper.

Because of copyright issues we are not able to include much more figures to illustrate the state of the art. We changed some of them to clarify the presentation. We also added as suggested tables to help the reader's understanding.

Thank you again for your comments.
Best regards
D. Vivet

Reviewer 2 Report

This paper provides a comprehensive survey on Visual-LIDAR
SLAM. After a sum up of the basic idea of SLAM and its implementation, 

Why the formula is not defined? Most of the formula is not defined.

I do not think you require to add a new section for abbreviations?

In Fig 1 the author mentioned add new features? what is the new features?

The paper require extensive proof-reading

Some of the figure is not clear, it is good to draw them again. For example Fig 1.

Author Response

Thank you for you review. Our answers in the text :

Why the formula is not defined? Most of the formula is not defined.

We have now defined all the equations, thank you for noticing this

I do not think you require to add a new section for abbreviations?

As suggested, and because all the abbreviations are defined in the text, we removed the optional section of the template

In Fig 1 the author mentioned add new features? what is the new features?

The figure 1 has been modified to be clearer. New features are in fact new landmarks. It represents the observed elements that are not yet in the map.

The paper require extensive proof-reading

The paper has been corrected by a native english speaker

Some of the figure is not clear, it is good to draw them again. For example Fig 1.

We modified the figure as requested. We hope the new versions are much more informative.

Thank you again for your comments.

Best regards

D. Vivet

Reviewer 3 Report

This document is of a scientific and original nature related to the simultaneous localization and mapping.

This document provides a survey on Visual-LIDAR SLAM - review on the state of the art of SLAM research.

For a better clarification, please edit your paper as follows:

* Figure 3 - text in the image is unreadable - printed on lines.
* Figure 4 - text on the x is unreadable - missing line.
* Figure 6 - text on the image is unreadable.
* Figure 7, 8 - text on the images are too big.
* Figure 9 - text in the image is unenhanced
* To adapt literature to a template.
* The equations in Chapter 2.1 are not numbered.
* There is no explanation for some variables below equations

Please edit the post according to previous comments and
after minor changes I recommend the publication.

Author Response

Thank you very much for your review. Please find our answers in the text :
* Figure 3 - text in the image is unreadable - printed on lines.
* Figure 4 - text on the x is unreadable - missing line.
* Figure 6 - text on the image is unreadable.
* Figure 7, 8 - text on the images are too big.
* Figure 9 - text in the image is unenhanced

Thank you for your comments about the figures. We have modified/adapted all the figures to have clearer legends and text.

* The equations in Chapter 2.1 are not numbered.

Correction done, all equations are now numbered.

* There is no explanation for some variables below equations

Correction done, all variables are now explained.

* To adapt literature to a template.

We have used the provided latex journal template for our literature. We also have completed missing information for some references.

Thank you again for your comments

Best regards

D. Vivet